# Improving Outcomes for Regional Families in the Early Years: Increasing Access to Child and Family Health Services for Regional Australia

**DOI:** 10.3390/ijerph21060728

**Published:** 2024-06-04

**Authors:** Jessica Appleton, Deborah A. Stockton, Marie Dickinson, Deborah Debono

**Affiliations:** 1Tresillian Family Care Centres, Mackenzie St., Belmore, NSW 2192, Australia; 2Faculty of Health, University of Technology Sydney, Ultimo, NSW 2007, Australiadeborah.debono@uts.edu.au (D.D.)

**Keywords:** child and family health (CFH) services, early childhood development, first 2000 days, regional and rural health, mixed methods, family-centred care, health consumer experience, healthcare access

## Abstract

Providing child and family health (CFH) services that meet the needs of young children and their families is important for a child’s early experiences, development and lifelong health and well-being. In Australia, families living in regional and rural areas have historically had limited access to specialist CFH services. In 2019, five new specialist CFH services were established in regional areas of New South Wales, Australia. The purpose of this study is to understand the regional families’ perceptions and experiences of these new CFH services. A convergent mixed-methods design involving a survey and semi-structured interviews with parents who had used the service was used for this study. Data collected include demographics, reasons for engaging with the service, perception, and experience of the service, including if the service provided was family centred. Triangulation of the quantitative and qualitative analysis uncovered three main findings: (i) The regional location of the service reduced the burden on families to access support for their needs; (ii) providing a service that is family-centred is important to achieve positive outcomes; and (iii) providing a service that is family-centred advances the local reputation of the service, enabling a greater reach into the community. Providing local specialist CFH services reduces the burden on families and has positive outcomes; however, providing services that are family-centred is key.

## 1. Introduction

An infant’s or young child’s brain is rapidly developing from birth to 3 years [1,2]. This rapid, critical brain development is foundational for ongoing health and well-being. A child’s early experiences and environment are fundamental to this brain development; children who experience a warm, attentive, consistent environment have more optimal outcomes [1,2]. For example, memories of warmth and safety from early childhood are related to a lowered risk of experiencing depression and anxiety during adolescence [3]. Conversely, infants are at risk of poorer outcomes when cared for by a caregiver who is emotionally unresponsive, unpredictable or inconsistent [4]. The substantial impact of a child’s earliest environment and experience on their health and well-being cannot be overstated; it is a potent determinate of psychosocial outcomes and has been shown to be more impactful on outcomes than some individual infant risk factors such as prematurity [2].

A key function of child and family health practitioners and services is screening, prevention, and intervention to promote the most optimal early environment. For example, in Australia, the focus of child and family health services on well-child health is reflected in the suite of community-based service activities, from surveillance and assessment of child development to psychosocial assessment of perinatal emotional well-being and interventions to enhance the parent–child relationship, including parent education and support programs [5]. Another key function of child and family health practitioners and services is communicating the importance of experiences during the early years of a child’s life for brain development. An example of this type of promotion is the New South Wales (NSW) Ministry of Health First 2000 Days Implementation Strategy, which raises awareness of the importance of the first 2000 days for long-term health, development, and well-being across the life-course of the child [6].

The NSW First 2000 Days Framework outlines three objectives, the third being “… additional services for those who need specialised help, when they need it” (p. 9) [6]. Families in regional and rural areas have access to primary-level child and family health services providing vital universal child health services; however, in many cases, such services, where they are in place, do not have the resources to provide a more intensive level of service to respond to families requiring additional support to address complex early parenting challenges often exacerbated by risk factors and vulnerabilities [7]. Such risk factors and vulnerabilities can contribute to adverse childhood experiences that can have a cumulative effect with implications for intergenerational risk for families. To ameliorate such risks, effective support or intervention to enhance protective factors is required [8,9].

For families in regional and rural communities, access to the stepped-up care of specialist child and family health practitioners and services is often less accessible due to geographical distances compounded by socio-economic disadvantage (cost of travel), sleep deprivation and the psychological co-morbidities often associated with early parenting difficulties [7,10]. These issues are not unique to Australia. Availability of services, geographic location, and capacity of services are known barriers to accessing health systems in many countries [11,12]. The importance of providing access to health services for children and families living in rural communities has been emphasised by the United Nations High Commissioner for Human Rights, who identified that children living in rural communities are at risk of being left behind [13]. Likewise, the Australian Health Ministers’ Advisory Council in the National Framework for Child and Family Health Services identified children residing in rural and regional communities as being at particular risk of experiencing vulnerabilities that can have repercussions for their health and being across the life course [14].

In 2019, five new specialist child and family health services were established in rural and regional areas of New South Wales (NSW), Australia. These services were an extension of a primarily metropolitan-based specialist child and family health organisation, The Royal Society for the Welfare of Mothers and Babies, known as Tresillian Family Care Centres (Tresillian). As these were five of only, at the time, seven regional Tresillian centres in NSW, research was conducted to understand the regional family’s perception and experience of these new centres. Understanding family perception is key for initial and ongoing engagement with child and family community health services [15,16]. This research therefore had the following research aims: (i) to investigate local parent perceptions and experiences of the Tresillian Regional Family Care Centres (TRFCC), including their perceptions of what would be provided by the service and if this was influential on accessing the TRFCC; (ii) to understand if and how the service enabled positive outcomes for parents related to their parenting and parent–child relationships; (iii) to determine the perceived role of the TRFCC within the local community; and (iv) to determine how the TRFCC supported links to local services or networks.

## 2. Materials and Methods

### 2.1. Study Design

A convergent mixed-methods design was used, involving a quantitative phase, qualitative phase (Figure 1) [17]. This approach involves separate collection and analysis of the quantitative and qualitative data, followed by triangulation of the quantitative and qualitative findings [17,18]. Triangulation involves side-by-side comparison of findings, inspecting for convergence or divergence [17]. The triangulation of the quantitative and qualitative findings produces meta-inferences, which are shown in a joint display and weaving narrative approach [17,18]. A joint display is a means of visually demonstrating the side-by-side comparison, and the weaving approach to integration is when the quantitative and qualitative findings are shown together on a concept-by-concept basis [18]. Both the quantitative (QUAN) and qualitative (QUAL) phases are treated with equally importance in this design [17].

### 2.2. Study Setting

In the state of NSW, there are fifteen local health districts (LHDs) that provide publicly funded hospital and community-based health-care services. Six LHDs cover metropolitan areas, and nine cover regional and rural areas. The TRFCCs in this study were established in partnership with five of these LHDs: Southern NSW, Western NSW, Far West, Hunter New England, and Mid North Coast. These five LHDs comprise major regional centres, smaller rural towns, and remote communities across both coastal and in-land contexts [19]. A framework for data-driven service development decision making was used to identify and prioritise locations for the centres, incorporating four key domains: Population, health outcomes, socio-economic indexes for areas (SEIFA), and existing coverage [20]. The framework (developed by Tresillian in collaboration with 180 Degrees Consulting) was specifically for the purpose of informing Tresillian regional and rural service development. The framework draws on data from the four domains noted above to create an index indicating communities’ level of need based on population size, vulnerabilities and existing child and family health service coverage. The use of this tool facilitated the collation of data regarding the suitability of locations for Tresillian services, which was tested further through consultation with local stakeholders and funding bodies to inform evidence-based service development decisions.

### 2.3. Participants

Clients who had attended a service from any of the five TRFCCs were invited to participate. Invitations were delivered in three routes: (1) Clients were invited via email after their appointments at the service; (2) posters advertising the study were placed in the TRFCC waiting rooms; and (3) the study was advertised on the organisation’s social media sites. The invitation directed clients to a landing page of the study hosted on an online research survey tool. This landing page had information about the study, including a downloadable participant information sheet. Clients could then check their eligibility, provide their consent and progress to the completion of a survey. On completion of the survey, participants could express their interest in participating in a follow-up interview by providing their contact details [18]. A stratified, purposeful sample was then used to interview clients from each of the five service locations.

### 2.4. Data Collection

Following consent, participants completed the online survey (phase 1). This survey gathered the centre location, basic demographics about the participant and their family, including their age, relationship status, education, country of birth, primary language, and number of children. Participants feelings of support generally from their individual support network were ascertained through three questions used in previous research with a similar cohort of parents [21]. These three questions asked about the support they received from their partner, and family/friends (response options: “I get enough help”, “I don’t get enough help”, “I don’t get any help”, “I don’t need any help”), and how often they need support but cannot find this support (response options: “very often”, “often”, “sometimes”, “never”, “I don’t need it”).

The survey collected the main reasons for the participant’s visit to the TRFCC, how they were referred and what they would have done if this service were not available. The survey asked about participant perceptions of the TRFCC, including both closed and open-ended questions. They were asked if they thought the centre would be useful for parents in their community (response options: “yes”, “no”, “maybe”), with the option to elaborate on their response with open text. They were asked if they had any expectations prior to accessing the service (response: “yes”, “no”) and if these expectations were met (response: “yes”, “no”), again with the option to elaborate on their response with open text. There was also a service checklist about services they were referred to or encouraged to access. Participants identified the services they were referred to or encouraged to access and then responded if they had attended, planned to attend or would not attend the service.

The survey questions also explored participants’ experiences with the service, including a measure of family-centred care, the family-centred care domain of the Promoting Healthy Development Survey [22].The family-centred care scale includes 8 items (see Appendix A) ranging from 1 (never) to 5 (always). Average scores were calculated. The survey also asked three questions on their perception of the impact of attending the service; these were answered “yes”, “no” or “unsure”. The questions asked if the service supported them in (1) meeting their family’s goals (goals), (2) changing their relationship with their child (relationship), and (3) changing their confidence as a parent (confidence). In addition to the “yes”, “no” or “unsure” closed responses, participants had the option to elaborate on their responses with open text.

For phase 2, a semi-structured interview was conducted. Interviews were audio recorded and transcribed verbatim. Member checking was conducted by emailing deidentified transcriptions back to participants.

### 2.5. Data Analysis

Analysis of each phase was conducted separately, guided by the research aims. Analysis for phase 1 involved descriptive statistics used to describe the sample and the responses to the survey questions. Non-parametric tests (Kruskal–Wallis or Mann–Whitney) and Pearson’s chi-square were used to test for differences between participant characteristics and expectations and the family-centred care scale, goals, relationship, and confidence questions. All statistics were run using SPSS version 28 [23]. All open-text responses in the survey were coded using inductive content analysis, with the unit of analysis being each participant’s response [24]. The procedure involved, firstly, open coding of the response, followed by grouping of the codes, and lastly, categorization. One researcher performed the initial coding, with a second researcher reviewing the coding. If there was disagreement on coding, this was discussed, and consensus was reached.

Analysis for phase 2 involved inductive thematic analysis of the interview transcripts. This is a six-step process of familiarisation, initial coding (both descriptive and interpretive codes), formation of themes, cross-checking themes, naming themes and finally, reporting the themes [25]. Two researchers completed this six-step process each on half the transcripts. Consistent with the process in phase 1, any differences in coding or developing themes were discussed and consensus reached.

The findings from phase 1 and phase 2 were compared and merged through a process of triangulation [17]. This process involves side-by-side comparisons, considering how the findings from each phase either converge or diverge to answer each of the research aims. All members of the research team met to review and discuss the emergent themes and sub-themes to ensure the terminology accurately represented the findings.

### 2.6. Ethics

This study was conducted in accordance with Australian and international codes of ethical conduct in human research and was approved by Sydney Local Health District—Royal Prince Alfred Human Ethics Review Committee protocol number X20-0549 & 2020/ETH03167. All subjects gave their informed consent for inclusion before they participated in the study. Participant consent was completed electronically at the start of the survey (phase 1) and verbally at the start of the interview (phase 2).

## 3. Results

A total of 148 potential participants responded to the recruitment upon arriving at the survey landing page. Of these, 82 confirmed their consent and commenced the survey; of these, 14 were excluded. These 14 were excluded as they answered zero (*n* = 5) or only one question (*n* = 8), or were duplicate entries (*n* = 1). The final sample size for phase 1 was *n* = 68. There was a total of 4.6% missing data, with no difference in missing data across the five locations; therefore, an available-case analysis was conducted. For phase 2, *n* = 21 participants who completed the survey expressed interest in completing an interview. Of these, *n* = 10 were interviewed. Participant characteristics and the services provided are summarised in Table 1.

Participants were from each of the five locations; however, there were more (39.7%) from Western NSW than from other sites, and there was only one participant from the Far West. All but one of the participants nominated their role as a mother (80%), the average age of participants was 31.9 years, and the average age of the child they attended the service with was 8.3 months. Most participants were in a married or long-term relationship (88.2%), and many attained tertiary-level education (60.3%). Half were on leave from paid work, and most were born in Australia (83.3%) and spoke English as their primary language (88.2%). Many participants had only one child (60.3%) and had accessed the service via a single modality (69.1%), and the main reason for accessing the service was sleep and setting related (69.1%). Regarding participants feelings of support from their individual support network, 66.2% (*n* = 45) felt they received enough support from their partner, and 47% (*n* = 32) felt they received enough help (or did not need help) from their family and friends. However, 73% (*n* = 50) felt they sometimes, often, or very often needed support or help but could not get it from anyone.

Triangulation of the quantitative and qualitative analyses uncovered three meta-inferences (Table 2). These will now be described in detail.

### 3.1. Regional Locations, Reducing the Burden on Families and Increasing Access to Support

If the local TRFCC was not available, only 14.7% (*n* = 10) of participants stated they would have travelled to attend a similar metropolitan-based service. Depending on the site, the closest metropolitan-based service was either across a state border or up to 1000 km away. It is therefore unsurprising that almost half (*n* = 31, 45.6%) reported they would not have accessed this type of service or selected “other” (*n* = 3), with one participant indicating they might have tried to access online services (ID 110) and another sharing they would have *“probably ceased breastfeeding” (ID 130)* as they believed they had exhausted all the available support options. The remaining participants (*n* = 24, 35.3%) reported they would have attempted to find another local service to meet their needs. But few (*n* = 6, 8.8%) knew of any other similar services locally available.

This is supported by the qualitative findings in the theme, Increased access is much more than just geography, where participants described that they would be left with the burden of trying to find a service to access the support they needed.


*I don’t know what I would’ve done … … I’d probably have to do a lot of … I would have had to probably jump through a lot more hoops, saying, “Oh, it’s affecting me mentally,” or you know, “It’s affecting her health.”*

*ID 62*


*So I probably would’ve contacted a sleep consultant online and I don’t know what I would’ve done about her feeding. I probably would’ve just given up* [breastfeeding] *and tried to bottle feed her.*
*ID 119*


This burden is related to both time and resources for the parent as well as the time and resources of other health professionals who are not necessarily able to provide the support required. The TRFCC is an additional resource for primary healthcare services, such as primary care child and family health nurses and GPs (general practice physicians). The TRFCC was identified by participants as a step-up service delivery level for those families requiring additional, more intensive support than the primary healthcare services had the capacity to provide. The services also provided a local referral pathway from the primary level to a specialist secondary level service, which was particularly welcome in areas with limited support.

*We’ve had a lot of issues because I‘ve got a tiny baby who, a lot of the standard* [child and family health nurses] *were freaking out about.*
*ID 10*



*… [town participant lives in], where I am is remote. Oh, no. Sorry. I think it’s classified as rural, but we’re the most trendy town in Australia at the moment … … There’s so many new people … … which is amazing for this town … …. but support wise, it’s abysmal.*

*ID 141*


Importantly, the increased access meant that participants received the right service at the right time. Participants reflected that the support from the TRFCC was received at just the right time to prevent their situation from deteriorating.

*I dread to think what I would’ve done actually. How bad it was because I’d already tried* [different service] *… I think the Sustained Home-visiting* [Extended home visiting service available from the TRFCC] *really was what saved us. So I don’t know what I would’ve done because we were a real mess when we came out … out … we’d just come out of two weeks of quarantine* [COVID related quarantine].
*ID 10*


[What would have happened if the TRFCC was not available?] *Honestly, I don’t know. I think I would’ve really, really struggled. I wouldn’t have known about resources like PANDA* [Perinatal Anxiety & Depression Australia], *would’ve had issues with a sleeping bub, wouldn’t have known about the silent reflux. I probably wouldn’t have been doing too great if I’m honest.*
*ID 136*


Because there often were not local accessible services that provided a similar service, the TRFCC participants expressed feeling “lucky” that the service was available in their local community:


*‘invaluable’ is a word I would use … … I’m just very grateful to them … … And honestly, I wish there were more centers out further west because I’m sure a lot of people could use their services and benefit from it greatly. I’m just really lucky that I’m half hour, 40 min from town, so it wasn’t such a big thing to get in there.*

*ID 136*


While increased access was much more than just geography, as the last quote testifies, it was also about geography and the distances families would otherwise have had to travel if a local service was not available. It was clear that having a local service that was physically close to the participant’s home increased access to and awareness of this type of service.


*Having something up here, you know, makes it so much easier. Yeah, but I think too, if we didn’t have Tresillian in town, I possibly wouldn’t have even really been overly aware of the service.*

*ID 119*


The TRFCC service also referred clients to other services to meet their multifaceted needs. Most participants (*n* = 50, 73.5%) were referred to, or encouraged to access, other services. This included referring back to primary child and family health clinicians, participating in local playgroups and making referrals to Tresillian residential services, which are primarily metropolitan based. Qualitative findings also identified that the TRFCC facilitated connections to services and resources otherwise unknown to the parent, including other local services and tertiary-level residential services when required.

*I think I had a bit of a blocked duct, so she referred me to go see a lactation consultant at Child and Family Health Services, just in relation to trying to get that sorted … … They also just pass along resources and things for PANDA* [Perinatal Anxiety & Depression Australia]
*ID 136*


### 3.2. The Importance of Family-Centred Care for Positive Outcomes

Participants reported their experience of the service they received at the TRFCC on an eight-item family-cantered care scale. Based on this scale, most participants received a service that was family-centred in nature, with a mean of 4.48 and a median of 4.75 (out of 5). The majority of participants (82.4%, *n* = 56) reported the service assisted in working towards their goal to address the main reason for attending the service. Similarly, a majority felt their confidence had changed (72.1%, *n* = 49) and their relationship with their child had changed (50%, *n* = 34) because of attending the TRFCC service. These *positive outcomes* were also identified in the qualitative phase. Participants described these positive outcomes, including the parent’s improved responsiveness to their child and improved confidence:


*… it was like almost getting a translational, like ‘learning to speak the baby’s language’. So, learning all their visual, non-verbal cues like … have a little bit more understanding of what your baby was trying to tell you. And yeah, so it made you feel a lot less lost.*

*ID 29*



*And so [nurse name], the nurse at the time helped me settle her like as a bit of a demonstration and I was just like, ‘oh my goodness, this actually works’. Because when you read the online or in a book, you’re like, ‘there’s no way that would work for my baby’. But then actually doing it with the support was really helpful. And that’s what gave me the confidence to do it myself.*

*ID 7*


Nursing staff working within a family-centred framework was important in facilitating these positive outcomes, as there was a significant difference in the family-centred care score between those who scored lower on the family-centred scale and those who reported that “no” their confidence as a parent had not changed or “no” the service did not help them meet their goals (Table 3). This is supported by the qualitative findings in the theme, *family-centred care*, where participants described a flexible, individualised service where they felt reassured, listened to, understood, reassured, and were given practical expert advice.


*I got support, which meant I felt like I was confident to keep doing the techniques that I was doing to get to a specific point instead of just not bothering. And I felt I was being heard and helped instead of just pushed aside and told this is normal, you’re not sleeping because you have a small baby and just get on with life. It was an acknowledgement that, “Cool there are things that we can do to help, let’s start working on it. And are you comfortable with these or are you not comfortable? Have you tried this? Let’s try something else.” It was the right level of support.*

*ID 141*



*Every time I walked away having learned something and it was beneficial.*

*ID 136*


This indicates that when nurses provide care for a family, there must be a focus on doing so in a family-centred way to effectively support parents to meet their parenting goals and be empowered to feel more confident. Conversely, those few who had negative experiences explained that the advice or approach provided was not a good fit for themselves and/or their infant, and they were not given other options. This lack of flexibility would not be considered family-centred care.


*Very ‘textbook’ approach. It feels as if the information provided is a one size fits all approach and all babies are different.*

*ID 21*


### 3.3. The Importance of Family-Centred Care for the Role and Reputation of Regional Services as an Enabler for Reaching Parents and Communities

Participants were asked about their expectations of the service and if it met these expectations. Before their first appointment with the service, most participants (*n* = 40, 58.8%) did not have any expectations. Those who had positive expectations described expecting supportive advice or help for infant behaviours (e.g., sleep) and parenting support. Some had a negative expectation that the approach might be authoritative and inflexible and were surprised by the family-centred approach, tailoring care to the needs of the family.

*I was hesitant to engage with* [TRFCC] *as I didn’t know what their approach was towards sleep and settling. I didn’t want to engage with a service that forced me to leave my baby crying for long periods of time, this is kind of what I expected and prepared myself for. My expectation above was not met thankfully. Tresillian has a lovely approach towards sleep and settling, encouraging attachment and care for your baby at all times. I was quickly made aware of the active and passive ways of comforting my child, none of which were to leave her crying for long periods of time.*
*ID 20*


Over half the participants (*n* = 38, 55.9%) reported that the service met their expectations; the others either had no expectations (*n* = 24, 35.3%) or reported the service did not meet their expectations (*n* = 6, 8.8%). Two of these six had negative expectation, as noted above, so they were pleased that the service was not what they expected.

There was a relationship between the service meeting a participant’s expectations and the family-centred scale score (H(2) = 8.8, *p* = 0.012). Pairwise comparisons show a significant difference between those who said the service met their expectations (*n* = 36, FCC score mean = 4.75) and those who said the service did not meet their expectations (*n* = 6, FCC score = 3.08). This is supported by the qualitative findings in the theme, family-centred care, demonstrating that those participants who described feeling heard, understood, reassured, and supported with a friendly, nurturing, flexible, individual approach providing practical, expert advice reported a positive experience.


*The staff at Tresillian were open and understanding, not judgemental at all and willing to help out every step of the way with also carefully considering a mothers mental health.*

*ID 145*



*That was really good just to hear it from someone else just saying, “You’re doing everything you can. I can see that you’re trying to get into a good routine. You’re doing the right things.” That reassurance was really valuable.*

*ID 37*


On the other hand, for those few where their expectations were unmet and they reported a negative experience, this was often related to an approach that was not flexible and therefore not family centred.


*I think that was the only option given at the time. And, they only had particular times that I could go. I remember that being an issue … So that was really quite hard. I remember when I had the appointment, it was basically not her sleep time so there was no point in trying to settle her because … Yeah. So we just spoke.*

*ID 31*


Parents’ experiences at the service are important for the ongoing work of the TRFCC, as positive experiences encourage parents to promote and recommend the service to other local families.


*I think it’s a truly brilliant service and definitely one that we will continue to use and support within the area and recommend onto other people.*

*ID 119*



*So anyone that asks me about it, I tell them, “Yeah, make sure you get down to Tresillian if you need the help.*

*ID 29*


Most participants heard about the local TRFCC from other local child and family health services (*n* = 52, 76.5%) or word of mouth (*n* = 12, 17.6%). Prior to this, most had some awareness of Tresillian as an organisation (*n* = 47, 69.1%). This is supported by qualitative findings where participants described being referred to or made aware of the TRFCC by other local clinicians. However, qualitative findings also included recommendations from participants to continue to increase awareness of the local TRFCC through promotion and providing more services or resources to continue to meet the needs of the local community.


*… it would be great if more health professionals in [town participant lives in], like more local people to me, knew about the actual services that Tresillian has and the actual things.*

*ID 141*



*I think my only thing would be if that they could be better supported in, you know, having more nurses that can work from the centre so that they can and actually reach more people.*

*ID 119*


## 4. Discussion

Access to specialist child and family health services is important for supporting optimal early environments for children to thrive [1], including addressing psychosocial vulnerabilities and psychological morbidities such as parental perinatal health concerns [26]. Indeed, intervening in the early years of a child’s life to identify and address risks through evidence-based interventions has been shown to effectively prevent or reduce health issues later in life [27,28,29]. This study demonstrated the impact specialist child and family health practitioners and services can have for families living in regional and rural areas, which is vital given the inequities in health outcomes between those living in rural communities and their counterparts residing in metropolitan areas [30,31]. These services reduce the burden (distance, time, and financial cost) on families and increase access to timely, effective support. In so doing, the TRFCCs are addressing priorities identified by rural parents and health service providers, including “equity of access, flexibility and timely response to prevent challenges from escalating to crisis point” (p. 229) [10]. This study showed that the services can have positive outcomes in terms of early intervention and parenting confidence, however, providing services that are family centred is key. These services are valued within the community, and parents express the need for the continued expansion of services to reach their communities.

These TRFCCs provided a local service to rural and regional families that was not available previously. Many of the parents may not have accessed this type of service if the TRFCC was not available locally. Implications of the lack of accessibility to health services include the misapplication of other health services, for example, the use of emergency services for primary healthcare [12]. Other implications of the lack of accessibility to health services include forsaking seeking help and the deterioration of a situation due to a delay in accessing the right services [7,12]. The health outcome gaps between metropolitan and rural populations have highlighted the impact of inequities in access to health services [31]. Even when there is no cost charged for access to a health service, socio-economic disadvantage in rural areas can further impact access due to the cost and time required to travel to services away from the local community [32].

The findings indicated that the TRFCC mitigated this by providing early intervention, with some parents expressing they did not know how they might have managed their early parenting challenge if the TRFCC was not available in their community. The TRFCC provided the right service at the right time and was able to provide care coordination through recommendations and referrals where other assistance was needed, supporting a “no wrong door” approach. The provision of this care coordination is a key support for parents that can bolster parent and family resilience when facing challenges [28]. This care coordination and a no wrong door approach to required services are vital in rural and regional areas where service availability is limited.

Importantly, parents shared that accessing the service was related to positive outcomes. These positive outcomes included reaching their own parenting goals and increasing parenting confidence. Supporting parents in the early years is important due to the impact a positive, warm, attentive, and consistent environment can have on a child’s health outcomes in the early years and beyond [1,2]. It is argued that caregiver regulation is a key element in providing a consistent, supportive early environment for optimal infant growth and development [4]. When a caregiver can help an infant regulate their emotions when experiencing a distressing situation, this structures the infant’s recovery from this distress [4].

Seeking to increase parental confidence and capacity to respond to the needs of their child has been identified by parents and community stakeholders as a primary function and key priority for specialist child and family health services such as the TRFCCs [10]. Parenting confidence and parenting self-efficacy refer to and often measure a parent’s belief in their ability to perform in their role as a parent [33]. Increased parenting confidence or self-efficacy is associated with an increased parental sense of self-sufficiency and a capacity to problem solve parenting challenges. It is also associated with positive parent–child relationship outcomes, including when experiencing stressors such as maternal mental illness [34].

An important finding from this study is the importance of family-centred care in achieving positive outcomes for families. Family-centred care is delivered within the context of relationship and strengths-based approaches, which are key to contemporary service delivery when working with families in the early years [35,36]. Family-centred care involves taking a facilitative approach, working with parents to enable the co-production of new knowledge and skills, and thereby creating an environment of joint ownership of problem solving and decision making [37]. This includes working collaboratively with parents, so they take the lead in identifying goals that are meaningful for their family [27]. This approach sends a potentially powerful message to parents about valuing and building on their existing knowledge and skills, contributing to the building of parental confidence and self-efficacy [5].

Likewise, this study found a family-centred approach was important for the ongoing reach of the services into the community. It follows that a family-centred approach includes providing options to families as to other services with whom they could engage to address multifaceted challenges, with the child and family health practitioner taking a facilitative role in care coordination and navigation [10]. Given the often complex and multifaceted nature of challenges experienced by rural families with young children, it is vital that health service providers develop a strong understanding of the local community context, and collaborative working relationships with community-based services in their local communities to create a well-integrated local service system network [38,39,40].

Integration into the location service system network requires establishing, developing, and maintaining good working relationships between service providers and individual professionals. These services were planned and implemented with the understanding that these relationships are key [20]. This is supported by a collaborative approach that is receptive to both client, community, and stakeholder needs [39]. One way in which the TRFCCs were receptive to community needs was the establishment of partnerships with Gidget Foundation Australia to co-locate Gidget services within regional TRFCCs, providing local access to perinatal psychological supports in areas where this is notably absent [41]. Tresillian and LHD stakeholders involved in the service integration have shared that working in partnership involves working as a “team”, mutual professional capacity building, and clear referral pathways with closed-loop communication [42]. Integration of services has allowed for an enhanced experience for clients, as demonstrated by these findings, with many clients referred to locally available services.

The TRFCCs can continue to be receptive to the needs of regional and rural families through the recommendation to increase awareness and the provision of more services. This includes providing multiple modes of delivery to increase access to services for rural and regional families. The TRFCCs provide not only face-to-face services but also support for families via telehealth. Telehealth should not be seen as a replacement for local services but rather as an important adjunct, extending the reach of the services to families living in more remote, geographically isolated areas or who cannot attend the service in person for a range of reasons [43,44]. This is consistent with research that has demonstrated families require a suite of modes of service delivery options, providing choice while being delivered by healthcare practitioners who understand the local regional context and have established knowledge and relationships across the local service system network [10].

An independent economic analysis of the TRFCCs model using cost–benefit analysis (CBA) methodology was undertaken concurrently with this study to identify the financial, social, and economic costs and benefits attributable to the service [45]. The CBA study included a review of literature, analysis of pre- and post-clinical measures (including validated self-report measures of parental mental health (Edinburgh Postnatal Depression Scale [46]), parenting self-efficacy and confidence (Me as a Parent Scale [47]), and progress towards family goals (Goal Attainment Scale [48]), client consultation, and case studies. The program logic and literature review informing the CBA identified short- and long-term outcomes attributed to the TFRCCs, with outcomes valued within the analysis including improved child physical and social development [49], improved parental physical and mental health [50,51] and communities experiencing improved access to healthcare [52]. The CBA identified that for every AUD 1 invested, there was a close to three-fold benefit return of at least AUD 2.83.

It is critical to note that rural and regional communities are not homogenous but rather unique in terms of many factors, including culture, resources, challenges, and strengths. In response to the call for action by the World Health Organization [53] to scale up and adapt successful health service models for less-well-resourced settings, including rural communities, Stockton and colleagues [9] have conducted research that has informed the development of a framework for the collaborative co-design of specialist child and family health services specific to the needs of different community contexts. The resulting Framework for Collaborative Adaptation of Service Models for Child and Family Health in Diverse Settings (CASCADeS) and associated toolkit have been published [9]. The guiding principles centre on the importance of building trusting relationships between service planners and community stakeholders, taking the time required to build a comprehensive understanding of the local community context, and collaborating to review the service model to identify adaptations to ensure contextually relevant, appropriate, and effective service delivery for children and their families [9]. The framework is freely available to service planners, health practitioners, and community stakeholders at https://cascades.deborahstockton.com (accessed on 1 June 2024).

The findings of this study should be considered, noting the following strengths and limitations. The generalizability of this study is limited by a non-random sample of TFCC clients [54]; however, a strength of the mixed-methods sample integration is that the sample for qualitative phase 2 was purposefully drawn for diversity across the sites and was selected from the larger phase 1 sample supporting triangulation [17]. Related to the quantitative phase 1 sample, the size of the sample was small, which therefore impacted statistical power [55] and the generalisability of the findings, particularly to Far West LHD, where participation was minimal. Then again, a strength of the mixed-methods triangulation of quantitative and qualitative phases was that these findings supported the direction of the quantitative findings. The questions implemented to measure parenting goals, confidence, and parent–child relationships were simple and had good face validity but had not been tested for construct, content, or criterion validity. However, given that the aims were related to parent perceptions of the service, these simple questions were deemed appropriate.

A further limitation of the study is noted in terms of the cultural suitability of the services, recognising that 88.2% of respondents reported English as being their first language and 83.8% reported being born in Australia. As respondents were not asked to identify if they were of Aboriginal or Torres Strait Islander cultural background, this study was unable to explore considerations such as cultural safety and inclusiveness. It is recommended that future research on such service models include a strong focus on strategies to engage families within regional communities from different cultural backgrounds and the associated impact on the accessibility and effectiveness of child and family health services for these families.

Future research should assess the effectiveness of the services based on validated measures of specific relevant outcomes, for example, parenting competence or parental anxiety. Such research could leverage data collected through pre- and post-clinical intervention measures the organisation currently utilises for clinical assessment and monitoring purposes and extend data collection through a longitudinal study approach. The TRFCC should continue to evaluate their services as new elements are added or service provision modified. For example, since this study, there has been an increase in the provision of parenting group programs at the TRFCC sites. Finally, understanding if there is any difference in outcomes between metropolitan and regional sites will ensure families across locations have access to the best services possible.

## 5. Conclusions

Access to specialist services located within rural and regional communities reduces the burden for local families. This is particularly important for rural children and their families, who have been identified as being at higher risk of experiencing vulnerabilities exacerbated by geographical distance and socio-economic disadvantage that can impact the individual throughout their life course. This study has demonstrated that access to local healthcare practitioners, who work in a family-centred way to build a collaborative relationship with parents, is vital to providing early, effective child and family health service delivery, including health promotion and early intervention. By taking a family-centred approach, health service providers can address goals meaningful to the family while building parental confidence and improving outcomes for rural and regional children and their families.

## Figures and Tables

**Figure 1 ijerph-21-00728-f001:**
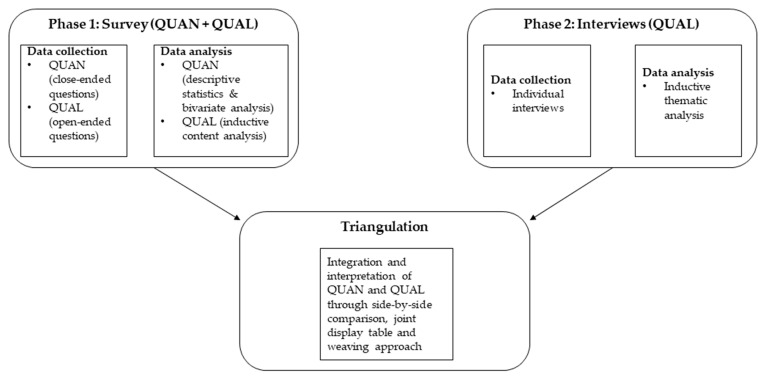
Study design. QUAN = quantitative; QUAL = qualitative.

**Table 1 ijerph-21-00728-t001:** Participant (*n* = 68) and service provided characteristics.

	*n* (%)	Missing *n* (%)
Location		0
Southern NSW	15 (22.1%)	
Western NSW	27 (39.7%)	
Far West	1 (1.5%)	
Hunter New England	12 (17.6%)	
Mid North Coast	13 (19.1%)	
Role		6 (8.8%)
Mother	61 (80%)	
Father	1 (1.5%)	
Age—parent	31.9 years (SD 4.7; range 22–43)	7 (10.2%)
Age—child	8.3 months (SD 6.8 months; range newborn—32 months)	0
Relationship		6 (8.8%)
Long term or married	60 (88.2%)	
Sole parent/carer	2 (2.9%)	
Education		6 (8.8%)
Postgraduate degree	16 (23.5%)	
Undergraduate degree	25 (36.8%)	
Certificate/diploma	12 (17.6%)	
High school	9 (13.2%)	
Current employment		6 (8.8%)
On leave from paid employment	34 (50%)	
Working full time	5 (7%)	
Working part time	14 (20.6%)	
Studying	2 (2.9%)	
Other	7 (10.2%)	
Country of birth		7 (10.2%)
Australia	57 (83.8%)	
Other	4 (5.9%)	
Main language at home		6 (8.8%)
English	60 (88.2%)	
Other	2 (2.9%)	
Number of children		6 (8.8%)
1	41 (60.3%)	
2	10 (14.7%)	
3+	11 (16.2%)	
Modality of visit ^1^		0
In-person	46 (67.6%)	
Home visits	18 (26.5%)	
Virtual	8 (11.8%)	
Group	16 (23.5%)	
Reason for visit		0
Sleep/settling	47 (69.1%)	
Feeding	9 (13.2%)	
Adjusting to being a new parent	6 (8.8%)	
My emotional well-being/mental health	1 (1.5%)	
Other ^2^	5 (7.4%)	

^1^ Could pick more than one option; ^2^ other: assistance with both feeding and sleep, baby had reflux and plagiocephaly, assistance with day sleep only, had twins or general education on caring of infants.

**Table 2 ijerph-21-00728-t002:** Joint display table triangulation of QUANT and QUAL.

Meta-Inferences	Interpretation	Quantitative Findings (Phase 1)	Qualitative Findings (Phase 2)
Regional locations, reducing the burden on families and increasing access to support	Having a local FCC reduces the burden on families when seeking access to this type of service. It also meant that services were provided in a timely manner preventing further deterioration of family functioning or maternal emotional health.	If the local service was not available, 45.6% would not have accessed this type of service; 35.3% would have tried to find a similar local service but only 8.8% knew of other similar local services. Only 14.7% would have travelled to attend a metropolitan-based service. 73.5% were referred/encouraged to access, other local or Tresillian services.	Theme: Increased access is much more than just geography
The importance of family-centred care for positive outcomes.	Most participants reported that their interactions with the TRFCC had positive outcomes related to their parenting. This was related to their perception of family-centred care at the TRFCC, nurses working within a family-centred framework supports parents to meet their parenting goals & increase their confidence. For those that had a negative experience at the TRFCC, elements related to family centred care were missing, such as flexible, individual service.	Goals: 83.8% “yes”, the appointments with the nurses at the FCC assisted in working towards their goal. Confidence: 72.1% “yes”, their confidence had changed because of attending the service Relationship: 50%, “yes”, there was change in their relationship with their child because of attending the service Difference in family centred care score related to goals, confidence and relationship.	Themes: Positive outcomes and family-centred care
The importance of family-centred care for the role and reputation of regional services as an enabler for reaching parents and communities.	Most participants did not have an expectation of the service before they attended and most found the service did meet any expectation and had positive experience. A minority had negative experiences which was related to a lack of family-centred care. Positive experiences promote the ongoing work of the TRFCC as parents recommend the service to other local parents.	55.9% reported that the service met their expectations and 8.8% reported the service did not meet their expectations. 35.3% did not have any expectation of the service. Family-centred scales score was related to the service meeting or not meeting these expectations. 76.5% heard about or were referred to the TRFCC from local child and family health services 17.6% had heard about the TRFCC through word of mouth	Themes: Family-centred care and increasing the reach in the local community

**Table 3 ijerph-21-00728-t003:** Family-centre care (FCC) score and goals, relationship, and confidence (*n* = 61) ^1^.

		Yes	No/Unsure	Mann–Whitney U Test
Assisted in working toward goal/s for accessing service	Proportion %, *n*	82.4%, *n* = 56	7.4%, *n* = 5	
	FCC score (median)	4.67 (4.82)	2.18 (2.13)	Z = 280, *p* < 0.001
Change in confidence as a parent	Proportion %, *n*	72.1%, *n* = 49	17.6%, *n* = 12	
	FCC score (median)	4.71 (4.88)	3.47 (3.44)	Z = 463.5, *p* = 0.001
Change in relationship with child	Proportion %, *n*	50%, *n* = 34	39.7%, *n* = 27	
	FCC score (median)	4.74 (4.94)	4.13 (4.63)	Z = 614, *p* = 0.020

^1^ Missing data 10.3%, *n* = 7.

## Data Availability

The data presented in this study are available on reasonable request from the corresponding author. The data are not publicly available due to privacy restrictions.

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
