# Peer review of "Improving Outcomes for Regional Families in the Early Years: Increasing Access to Child and Family Health Services for Regional Australia"

_ijerph, 2024, doi:10.3390/ijerph21060728_

Round 1
Reviewer 1 Report
Comments and Suggestions for Authors
The manuscript investigates the perceptions and experiences of regional families regarding newly established Child and Family Health (CFH) services in regional New South Wales, Australia.
I recommend the following before publishing:
- I suggest expanding the discussion on the scalability and sustainability of the CFH services. This could include a more detailed exploration of potential challenges and strategies to overcome them when replicating the model in other regional areas.
- Please expand on the framework used and the rationale behind its selection
- I suggest including the number of clients that were invited to participate in the methodology.
- Please consider elucidating the economic implications of implementing and maintaining such services in regional settings.
- While the study offers in-depth insights from New South Wales, the applicability of the results to other rural and regional settings globally might require further explanation.
- Please clarify the impact of sociocultural factors on the accessibility and effectiveness of CFH services.
- Consider exploring how different cultural backgrounds within the regional communities might affect engagement with these services.
- Please add more about the long-term outcomes of the CFH services.
- I suggest expanding the conclusion to offer a clearer roadmap for the practical application of the study’s findings. This could involve outlining steps that healthcare providers, community leaders, and other stakeholders can take to adopt the model discussed.
Author Response
"Please see the attachment."

Reviewer 2 Report
Comments and Suggestions for Authors
Thanks for offering me the opportunity of reviewing this paper, which explores the perceptions from families living in rural/regional areas in NSW (Australia) about the implementation of child and family health services in that areas. The study uses both quantitative and qualitative approaches. The focus of the study and approach are interest and relevant to the implementation of family services in rural areas. However, in my view the study could be better reported. Please find below key points that I think could improve the quality and chance of publishing this work.
Introduction: While it describes really well the need for the services in rural/regional areas, it does not say why it is important to explore family experiences. Thus a clear rationale for conducting this study, is missing.
Study aims should be reported at the end of the introduction not in the methods
The study design should be reported as the first subsection of the methods
What means “E” in the subheading “data collection”?
Data collection: provide more information of which basic demographic information was collected. Also, how data about ‘feelings of support’ was collected? Was this a questionnaire if so which type of questions? About perceptions of the TRFCC, were this open questions? What is the name of the measure about family-centred care? And how the participants reported their answers on the service supporting them (e.g., yes/no responses)?
Regarding the score for the family-centred care’s scale, it would read better the follow “ The family-centred care scale includes 8 items ranging from 1(never) to 5(always). Average scores were calculated”.
It is not possible to understand what means “the goals, relationship and confidence questions”. It is out of context.
Analysis plan.
It is unclear which study objective each analysis addresses. I suggest present the analysis plan according to the study objectives.
Results
In the first sentence, it is not clear what means“… at the landing page the study”.
Similar, to the analysis plan, I found difficult to understand how the results answer each study objective. In addition, reporting quantitative and qualitative results together impact the flow of the reading. My suggestion is to report quantitative and qualitative results separately.
Author Response
"Please see the attachment."

Round 2
Reviewer 2 Report
Comments and Suggestions for Authors
No further comments from me. The authors revised and commented about all the points I've raised. Well done!